# Brassinosteroids: An Innovative Compound Family That Could Affect the Growth, Ripening, Quality, and Postharvest Storage of Fleshy Fruits

**DOI:** 10.3390/plants13213082

**Published:** 2024-11-01

**Authors:** Fernando Garrido-Auñón, Jenifer Puente-Moreno, María E. García-Pastor, María Serrano, Daniel Valero

**Affiliations:** 1Department of Food Technology, Institute for Agri-Food and Agro-Environmental Research and Innovation (CIAGRO), University Miguel Hernández, Ctra. Beniel km. 3.2, Orihuela, 03312 Alicante, Spain; fgarrido@umh.es (F.G.-A.); jpuente@umh.es (J.P.-M.); 2Department of Applied Biology, Institute for Agri-Food and Agro-Environmental Research and Innovation (CIAGRO), University Miguel Hernández, Ctra. Beniel km. 3.2, Orihuela, 03312 Alicante, Spain; m.garciap@umh.es (M.E.G.-P.); m.serrano@umh.es (M.S.)

**Keywords:** 24-epibrassinolide, elicitors, crop performance, physio-chemical traits, secondary metabolism

## Abstract

Brassinosteroids (BRs), a new family of plant hormones, have been used in a range of food staples, oil crops, and cereals. However, the scientific literature pertaining to their use in fleshy fruits remains scarce. This review presents, for the first time, the knowledge developed over the last decade on the role of BR preharvest treatments in crop yield and fruit quality properties at harvest and during storage, although information about the effects of BR postharvest treatments is also addressed. This review revealed that 24-epibrassinolide is the most used BR analogue in research experiments, either as a pre- or postharvest treatment, with doses ranging from 0.1 to 15 μM. Additionally, most of the research has been conducted on non-climacteric fruit species. In most of these preharvest treatments, an increase in crop yield has been reported, as well as enhanced anthocyanin concentration in red-coloured fruit. In addition, increases in firmness, total soluble solids, and phenolic content have also been observed. On the other hand, BR postharvest treatments led to the maintenance of these fruit quality properties during storage due to increased antioxidant systems, either enzymatic or non-enzymatic ones. Finally, as future perspectives, it is proposed to extend the research about BR treatments to other climacteric fruits and to deepen the knowledge of how BRs regulate physiological aspects from preharvest to postharvest. Furthermore, it is essential to investigate the role of BRs in the prevention of rot and biotic stress.

## 1. Introduction

Throughout history, plants have developed sophisticated mechanisms in response to different types of stress, both biotic and abiotic, during their growth and development [1,2]. However, modern lifestyles and dietary habits contribute to global warming, with a detrimental impact on the planet. All of the reports from The Food and Agriculture Organization of the United Nations [3] alerts us that our current diet exceeds the recommended intake of fish, meat, or processed food while being poor in fruits and vegetables. Consequently, the global threat of climate change has a significant impact on agricultural practices, with varying effects on crop performance and fruit yield depending on the specific stressors encountered, including high temperatures, salt, heavy metals, and UV light, as well as the plant species in question [1,2,3,4]. In this sense, the recently developed omics-based strategies, including phenomics, genomics, proteomics, and transcriptomics approaches are aimed to enhance crop yield by developing crop species that are more tolerant under this climate change scenario [5,6].

Elicitors are molecules capable of inducing any type of defence in the plant and are produced in response to biotic and abiotic stressors. It can be said that the application of an elicitor acts on the plant with the same principle as vaccination; that is to say, the plant’s metabolism is activated, and it becomes more resistant to subsequent stress-inducing attacks. The use of elicitors has grown due to the benefits that are triggered by using them in crops, since they generally act as precursors of secondary metabolites, including phenolic compounds and phytoalexins. Plants can enhance their resilience to cope with the presence of these stressors at the cellular level, as well as activate different biochemical mechanisms. It is well known that the main consequence of these stressors is the production of reactive oxygen species (ROS), which generate harmful compounds that affect key physiological processes, such as photosynthesis, cell division and elongation, respiration, growth, and ripening [1,2,7]. As the process continues, the toxic effects can have a severe impact on crop yield, potentially leading to the death of the plant.

To mitigate the effects of abiotic stress, plants have evolved a range of adaptive responses, with the most significant among them being the production of plant hormones (PHs) and plant growth regulators (PGRs). These compounds have the capacity to enhance stress tolerance in plants. Nevertheless, in numerous instances, the attained endogenous concentration is insufficient to elicit the stress tolerance response [8]. For this reason, the preharvest application of these elicitors (PGRs and PHs) is a crucial and almost indispensable step. The five ‘classical’ PHs, namely auxins (AUXs), gibberellins (GAs), cytokinins (CKs), abscisic acid (ABA), and ethylene (ET), have been extensively studied in a wide range of horticultural crops [9,10]. However, in recent years, other compounds, which are also considered PHs, such as methyl jasmonate (MeJA), salicylic acid (SA) and its derivatives, including methyl salicylate (MeSA) and acetyl salicylic acid (ASA), melatonin (MEL), and γ-aminobutyric acid (GABA), have been used as potent elicitors against several abiotic stresses, with additional effects on crop yield and fruit quality [8,11,12,13,14,15,16,17].

In recent years, PHs and PGRs have been considered favourable eco-friendly alternatives due to their naturally occurring capacity to enhance tolerance against abiotic stress in many crops. Accordingly, past studies in the literature prove that there is a cross-talk among all these elicitors by the coordination of these signal transduction pathways [18,19]. Modern agriculture and horticulture are based on the scientific knowledge derived from the researchers’ outcomes, which can facilitate the development of new molecules that are able to modulate fruit growth. In this sense, brassinosteroids (BRs), a novel family of PGRs, which were discovered in the pollen of *Brassica napus*, have been used to enhance crop yield in a wide range of plant species, such as food staples, oil crops, cereals and fibre crops, mainly under stress conditions [20,21,22,23] and have been demonstrated to be effective in enhancing agronomic traits and stress resistance [24,25]. However, the scientific literature is very scarce in fleshy fruits. This review aims to provide an overview of the strong potential of BRs, as a naturally occurring tool, to be applied as preharvest or postharvest treatments by evaluating crop yield and fruit quality at the time of harvest or after postharvest storage. Most of the literature has been focused on climacteric fruit, while the knowledge of non-climacteric fruit has been poorly studied. Unlike other PHs, the use of BRs in agriculture is recognised as non-toxic, eco-friendly, and naturally occurring, particularly in the context of fruit production [26]. It is notable that BRs exhibit activity at concentrations that are extremely low, in the range of 1–100 µM.

## 2. Brassinosteroid Biosynthesis 

Brassinosteroid molecules are composed of four rings (A, B, C, and D) and a side chain (Figure 1). The formation of these molecules involves the condensation of blocks of five-carbon atoms, known as isoprenes. BRs are steroids with 27, 28, or 29 carbon atoms with different substituents in the A and B rings and in the side chain [4,27]. Chemically, more than 70 BRs have been identified from plant sources. Brassinolide (BL) is so far the one that produces the greatest biological activity among them and can be synthesised directly from campesterol (CS) or through the general synthesis of sterols. Plant sterols, in addition to their role as precursors of BRs, are integral components of cell membranes, where they regulate their fluidity and permeability.

As previously mentioned, the term ‘brassinosteroids’ was assigned to steroids that promote the growth of plant tissues. Nevertheless, several steroids are intermediates in their synthesis, which has led to uncertainty regarding the characteristics of BRs. To elucidate this situation, Bishop and Yokota [28] proposed defining BRs as steroids that have one oxygen at the C-3 carbon atom and additional oxygens at C-2, C-6, C-22, and C-23 (in accordance with the numerical sequence of the steroid carbons, as illustrated in Figure 1). BRs were initially isolated from the pollen of *Brassica napus* as a promoting compound of cell elongation [29]. They are regarded as the ‘sixth’ PH, with over 70 distinct BRs having been isolated and characterised. These BRs exhibit three primary structural forms: 5α-cholestane, 5α-ergostane, and 5α-stigmastan [30]. The different radicals present at rings A, B, and the side chain produce the BR family in which brassinolide (BL) is the most active BR, followed by catasterone (CS), the chemical structures of which are illustrated in Figure 2. Of all BR structures, 24-epiBL has been identified as the most active BR, exhibiting a wide range of physiological and metabolic effects and also being commercially available [31]. On the other hand, brassinazole has been found as a potent BR inhibitor. The biosynthesis pathways of BRs have been covered in excellent reviews, as well as the precursors, products, and inhibiting compounds [27,30].

The concentration of BRs in a particular plant tissue is largely dependent on local biosynthesis and degradation. The first steps of BR biosynthesis involve two pathways, the mevalonate pathway, starting from the combination of two acetyl-CoA molecules, and the independent mevalonate one, which starts from the combination of D-glyceraldehyde 3-phosphate and pyruvate, both rendering geranyl pyrophosphate. Then, the biosynthesis of C27, C28, and C29 BRs occurs throughout three complex pathways. In addition, BRs undergo different catabolic processes, including degradation, conjugation, and other modification processes. Thus, different glycosyl-transferases, reductases, acyl-transferases, and sulfo-transferases have been identified as important enzymatic activities acting directly on the most important BR compounds, BL and CS, and on their biosynthetic intermediates [27,30,31].

## 3. Brassinosteroids: A Key Plant Hormone for Regulating Plant Growth and Development

Brassinosteroids (BRs) play crucial roles in physiological processes, such as plant growth and development, as well as in plant responses to environmental stresses [4,23,24,31,32]. As with many other compounds that have been the subject of research, BRs have their own history [33]. The term ‘BRs’ was introduced by Mandava [34] to categorise plant growth-promoting steroids identified in the second elongation bioassay of beans. In the early 1990s, several Japanese chemist groups demonstrated encouraging advancement in the discovery of the biosynthetic pathway for BRs [35]. Concurrently, diverse groups of plant physiologists were engaged in research to elucidate whether these compounds constituted a novel class of plant hormones. However, it was not until 1996 that compelling evidence was found indicating their indispensable role in plant growth and development [36]. A brief summary of the history line of BRs is shown in Figure 3, starting in 1979 and delineating the main milestones.

## 4. Preharvest Application of Brassinosteroids in Fruit Crops

Since BRs were found ubiquitously along the plant kingdom, a vast number of BR responses at different levels, including genetical, physiological, biochemical, and molecular ones, have been reported. In plants, BRs are involved in seed germination, photomorphogenesis, photosynthesis, cell division and elongation, and fruit growth, among others [37]. Furthermore, BRs have been reported as a class of compounds with the capacity to safeguard plants from several abiotic (e.g., salinity, heat, flooding, drought, and heavy metals) and biotic (e.g., viruses, bacteria, and fungi) stresses [4,24,25,38]. BRs have been found in tree architecture distributing ubiquitously in all plant organs, including both vegetative (leaf, stem, and root) and reproductive (flower, seed, pollen, and fruit) structures. In general, BRs are found at very low concentrations, ranging from 0.01 to 100 μg kg^−1^ (fresh weight), depending on the plant organ and age. The highest levels are found in pollen and young growing organs, while BR concentrations decrease as maturation advances [23,37,39].

### 4.1. Fruit Growth, Development, and Ripening

From the botanical point of view, fruits are highly diverse, with the more complex ones being the fleshy fruits, but from a commercial standpoint, fruits provide us nutrients like vitamins, minerals, sugars, organic acids, and other non-nutrient constituents, mainly fibre and phytochemical compounds, which have health-beneficial effects. The biological cycle of the fruit encompasses different stages or phases: (a) the formation of the fruit, (b) fruit growth, which is divided into cell division and cell expansion, (c) fruit ripening, and (d) fruit senescence [40]. At the end of the cell expansion, the fruit has reached its maximum size, while during the subsequent ripening phase, the greatest changes associated with the organoleptic quality occur, including alterations in colour, an increase in sugars, a decrease in titratable acidity (TA), and a reduction in firmness (softening). Cell expansion is a critical requirement for cell growth and tissue formation in all plant organs. This process is regulated by the coordinated alteration of cell wall mechanical properties, biochemical processes, and gene expression. The primary wall of most cells is composed of cellulose microfibrils (sugar-like polymers) that are linked in a network to hemicellulose, which is embedded in a gelatinous matrix of pectins. For cell expansion to occur, the cell wall must undergo a brief relaxation or disruption of its interactions, accompanied by the addition of new cell wall components to prevent wall thinning and disruption. The final size of the fruit depends on many factors occurring during the development process, which could explain the observation that there are fruits of disparate sizes and with some discrepancies in shape within the same plant. Furthermore, additional factors at the genetic, edaphic, and environmental levels play a role, with genetic factors being the most significant.

The process of ripening is genetically programmed and involves a series of biochemical changes that modify the physical, chemical, nutritional, and functional properties of the fruit [40]. Those biochemical changes affect the taste and texture that a fruit undergoes during its last developmental phase. The primary roles of BRs are the regulation of both cell division and expansion [41]. However, the evidence substantiating the assertion that BRs regulate the growth and ripening of fleshy fruits remains scarce, although some evidence exists regarding the role of BRs in conjunction with other PHs (cross-talk) in enhancing the growth and quality of fruits [42]. As preharvest treatments, BRs have been used in a range of intensive crops, such as staple foods, oilseeds, cereals, and legumes [26]. On the contrary, the scientific literature on their use in fruit commodities is comparatively limited. Table 1 shows some examples of the application of BRs in some fruit species.

Furthermore, the application of 24-epiBL at a concentration of 10 µM has been demonstrated to enhance the number of flowers and fruits produced by pepper plants, resulting in an increased yield (fruit per plant) without any discernible impact on fruit weight and size [48]. In wine grapes, there is increased evidence suggesting that the ripening of grapes is related to the enhancement of their endogenous BR levels [49]. Subsequently, Babalık et al. [50] reported that preharvest 24-epiBL (0.6–0.8 mg L^−1^ at fruit set, veraison, and 30 days after) enhanced yield (bunch and berry weight), suggesting that the optimal time for application was at veraison since the antioxidant compounds (trans-resveratrol, β-carotene, and ascorbic acid) were largely enhanced. In red delicious apples, the foliar spray with 24-epiBL at µmo L^−1^ enhanced fruit quality trait (fruit size and weight, firmness, total soluble solids/titratable acidity (TSS/TA) ratio) and bioactive compounds (total anthocyanins and phenolics and total ascorbic acid with respect to control trees [51]. In strawberry plants, grown under water stress or normal irrigation, treatments with 24-epiBL at 0.1 mg L^−1^ promoted plant growth under the two regimes by increasing the number of leaves and chlorophyll content. In relation to fruit yield, increases in fruit weight (1.5-fold), number of fruits per plant (10%), and percentage of commercial strawberry quality were obtained after the foliar application of 24-epiBL [22]. Overall, the results suggested that BRs improved fruit performance not only under abiotic stress but also under optimal conditions.

With respect to fruit ripening, there is increased evidence that its regulation in non-climacteric fruits is due to ABA signalling [52]. Thus, in blueberries and sweet cherries, the accumulation of anthocyanins and the enhanced fruit colour are concomitant with ABA increase [53,54]. However, recent reports also claim a role of BRs. For instance, in the ‘Crimson Seedless’ table grape, the application of B60, a synthetic BR, at a concentration of 0.06 mg L^−1^ at the veraison stage, enhanced the skin colour at harvest [55]. In strawberries, the increase in BR endogenous content is responsible for directly controlling the ripening process, as was observed by Ayub et al. [56]. It has been suggested that glucose concentration during the ripening process could affect many steps in BR pathway biosynthesis [57]. Thus, BR treatment could be useful to increase skin colour in some red fruit species, including pomegranate, strawberry, and blood orange, that frequently fail to attain an appropriate and uniform colouration, and that have depreciated values at markets. On the contrary, in climacteric fruits, the promotion of ripening and stimulation of anthocyanin synthesis is mainly attributed to ethylene biosynthesis, without any observable enhancement in BR content, so BRs are supposed to not be involved in the process [58,59]. However, a recent report shows that the application of 24-epiBR (from 1 to 4 mg L^−1^) throughout the growth period resulted in an increase in peel anthocyanins in apple, a climacteric fruit [51].

### 4.2. Fruit Yield and Crop Performance

In the context of preharvest field applications, BRs induced a wide range of physiological processes, with a net impact on crop performance, especially in terms of enhancing crop yield. For instance, in Kyoho grapes, the treatment with homobrassinolide (HBL), applied three times during berry growth (before flowering, at full bloom, and before veraison) increased grape size [60] but also enhanced skin colouration and the content of total soluble solids (TSS), suggesting promotion of berry ripening [49]. In strawberries, three applications of BRs at 0.1, 0.2, and 0.3 ppm throughout fruit growth resulted in an increase in leaf number and weight, chlorophyll content, fruit size, and crop yield, irrespective of treatments, and the lowest doses advanced the onset of ripening [44], according to the increased TSS.

In a comparative study using three non-climacteric fruit species (blood orange, pomegranate, and two cultivars of sweet cherries), the preharvest application of 24-epiBL, at 0.01, 0.1, and 1µM, revealed different results depending on fruit species in relation to crop yield (Figure 4). The application of BRs at 0.01 and 0.1 µM significantly improved the yield of blood oranges (≘ 62 kg tree^−1^) compared to control trees (≘ 56 kg tree^−1^), while in pomegranate, only the doses of 24-epiBL at 0.01 µM enhanced crop yield (which was ≘ 80 and 72 kg tree^−1^ in treated and control trees, respectively). With respect to the two sweet cherry cultivars tested, an anomalous behaviour was observed. For both cultivars, the optimal dose was determined to be 0.1 µM, with production reaching ca. 20 and 30 kg tree^−1^ for ‘Sunburst’ and ‘Skeena’, respectively. It is noteworthy that the highest doses of 24-epiBL (0.1 and 1 µM) in ‘Sunburst’ had a better performance (≘ 22–18 kg tree^−1^) than control trees (≘ 12 kg tree^−1^). The explanation was that harvest time for ‘Sunburst’ is at the end of May, while the ‘Skeena’ cultivar is harvested at the end of June. According to the Spanish Meteorological Agency (https://www.aemet.es/ accessed on 2 January 2024), the province of Alicante had its wettest month of May since 1950, with rainfall of 195 litres per square metre. The results of this experiment demonstrate that 24-epiBL is an effective agent for improving crop yield, although the optimal dose and plant species may vary. Furthermore, 24-epiBL was able to mitigate the detrimental effects of heavy rain, a common abiotic stressor.

### 4.3. Fruit Quality Attributes

The concept of ‘quality’ is too complex to be studied in detail here. However, it can be noted that it refers to the ability to differentiate between two or more similar elements, with the one considered the ‘best’ being the result of this differentiation. In the case of fresh fruits, the quality components can be divided into several categories, including visual (size, colour, gloss, etc.), organoleptic (flavour, aroma, etc.), textural (juiciness, firmness, etc.), nutritional (sugar, proteins, lipids, etc.), functional (content of vitamins, minerals, phytochemicals, etc.), and safety (absence of natural toxins, pesticide residues, etc.). Certain markets also value other social quality components, such as the ‘organic’ origin of the products. The concept of quality is inherently subjective and, thus, a particular fruit species or cultivar may evoke different perceptions of quality contingent on the criteria by which it is valued. These may include considerations, such as productivity, resistance to pests and diseases, shelf life, and tolerance to mechanical damage, as seen through the lens of a farmer, marketer, or consumer, respectively. Nevertheless, it is possible to categorise the reduction in the quality of a product during postharvest handling as a distinct type of loss. This allows for a differentiation between quantitative losses, which are defined as losses of saleable products, and qualitative losses, which are characterised by a decrease in quality and, consequently, a reduction in price.

#### 4.3.1. Firmness, Total Soluble Solids (TSS), and Titratable Acidity (TA)

In non-climacteric fruit, the date of harvest is of great importance, as it allows for the selection of fruits that have reached an optimal ripening stage. Fruits that are less mature lack the organoleptic quality that consumers demand, including characteristics such as colour, taste, aroma, and flavour. During ripening, TSS increased, and TA decreased, both parameters being considered as good indicators of ripening. Since harvest and quality traits are specific for each fruit species and cultivar, these traits could be useful in determining the optimal harvest date for growers [40]. Nevertheless, the impact of BRs on fruit quality attributes remains poorly understood, despite a few previous studies (Table 2). These studies have demonstrated that foliar application of BRs can maintain fruit quality traits, such as firmness, TSS, and TA, during storage, thereby extending the shelf life. For instance, the firmness of persimmons was increased [61], as was that of ‘Red Delicious’ apples [51]. The application of 24epi-BL at concentrations of 0.3–6 μmol L^−1^ to ‘Thompson Seedless’ table grapes resulted in an increased content of TSS and TA [47]. Conversely, the BR treatment had no significant effect on ‘Red Globe’ or ‘Crimson Seedless’ grapes [55]. The application of a 0.1 ppm BR treatment to strawberries at the vegetative, flowering, and fruit set stages was found to result in an increased TSS/TA ratio [44]. The data suggest that BRs can enhance fruit consumption and overall quality by increasing TSS accumulation and decreasing TA content.

#### 4.3.2. Bioactive Compounds and Antioxidant Activity

Ascorbic acid, phenolic contents, and anthocyanins are considered important molecules having antioxidant capacity with health-beneficial effects. It is well known that fruit, in general, and red fruits particularly are potent antioxidants mainly due to the presence of ascorbic acid and phenolics, including anthocyanins. The role of BRs on these compounds has been poorly studied, although limited data exist. Table 3 shows some examples of the content of bioactive compounds and antioxidant activity in some fruit species at the time of harvest as affected by BR preharvest treatments.

Most of the reports have been focused on strawberries and wine grapes. Thus, EPL treatment increased anthocyanin content in Cabernet Sauvignon wine grapes [62] and Asghari and Rezaei-Rad [47] reported that 24-epiBL treatment to a vineyard, at veraison, induced higher content of total phenolic, antioxidant, and ascorbic acid levels in treated berries as compared with control ones. In addition, the activity of catalase and polyphenol oxidase enzymes was increased. Thus, BRs are able to enhance both bioactive compounds and antioxidant activity, with the final result of better fruit quality attributes. BRs show good potential for enhancing fruit phytonutrients, nutritional quality, and phytochemical contents, thus increasing the value-added properties of the crop.

## 5. Postharvest Application of Brassinosteroids

Fleshy fruits are highly appreciated by consumers and are part of the daily diet, as they are rich in dietary fibre, vitamins, and bioactive compounds related to health. The commercial value of fleshy fruits is affected by their texture, colour, taste, aroma, etc. Consumers are worried about their health and demand fruits and vegetables without chemical preservatives or non-environmentally friendly postharvest operations. However, fruits are very perishable during storage leading to quality losses, such as softening, decay, chilling injury (CI), colour changes, and senescence, with their rate of deterioration being dependent on the respiration rate, which in turn is affected by fruit species and the temperature of storage. The control of temperature is the most important factor for reducing the deterioration rate [63]. However, tropical and subtropical fruit species suffer from a CI physiological disorder when stored under low temperatures [40]. The main symptoms are dehydration, weight loss, accelerated respiration rate, pitting, and browning, among others.

The first target of CI is the damage of the cellular membranes and ROS production leading to an oxidative environment responsible for fruit deterioration. The membrane fluidity loss can be measured by the percentage of electrolyte leakage (EL). Also, the membrane changes from the liquid stage to a solid–liquid stage and the ratio between unsaturated fatty acids/saturated fatty acids (UFAs/SFAs) shows a decrease, and thus the bilayer of phospholipids becomes more saturated and less fluid [64,65]. Recent reports are based on the efficacy of several postharvest tools combined with refrigeration to alleviate CI and enhance shelf life, such as MeJa, MeSa, SA, GABA, MEL, and oxalic acid (OA), in pomegranate, yellow pitahaya, sweet cherry, and table grape [13,14,15,17,66]. BRs can also play a role in fruit CI tolerance during postharvest storage mainly in non-climacteric fruits such as zucchini [67]. BRs as an environmentally friendly safe regulator can be used not only for reducing postharvest losses by alleviating CI along with reducing decay but also for maintaining the nutritional quality of fruits and vegetables [51]. Zhu et al. [68] reported that the possible role of BR treatment in alleviating CI symptoms was due to increasing H_2_O_2_ accumulation. These authors confirmed that mandarin treated with BRs induced pathogenesis-related genes, like chitinase and phenylalanine ammonia lyase (PAL), concluding that the pathogen resistance by BRs is achieved via activating the systemic acquired resistance (SAR).

Apart from the well-known role of BRs in reducing CI damage, there is increasing evidence about their role in fruit ripening and quality. Moreover, the exogenous BR application would increase their endogenous content and obtain a response in fruit quality parameters. Some examples are shown in Table 4.

To be effective against CI, the molecule should demonstrate its role in increasing membrane integrity by lowering the activity of both lipoxygenase and phospholipase D, secondly, increasing the ratio of unsaturated/saturated fatty acids (UFSs/SFAs) and the antioxidant system activity, and finally, modifying the PAL and PPO enzyme activities [42,71]. Pomegranate fruits were treated with 0, 5, 10, and 15 μmol L^−1^ of 24-epiBL and stored for 84 days at 5 °C, and the results demonstrated that treatments alleviated CI, with the best dose being 15 μmol L^−1^ based on reduced weight loss, EL, fruit decay, and content of malondialdehyde (MDA). Also, antioxidant enzymes, phenylalanine ammonia lyase (PAL), peroxidase (POD), catalase (CAT), ascorbate peroxidase (APX), and superoxide dismutase (SOD) showed higher activity with respect to control fruit, although lower polyphenol oxidase (PPO) activity was observed, maintaining total phenolics [70]. In mandarin, postharvest treatment with 24-epiBL at 5 mg L^−1^ showed a 5-fold lower weight loss and 78% reduction in CI symptoms after storage for 50 days. Additionally, 24-epiBL enhanced H_2_O_2_ content, suggesting that H_2_O_2_ is responsible for increasing the shelf life [68].

## 6. Brassinosteroids and Human Health 

For centuries, natural products extracted from the plant kingdom have been used in folk and traditional medicine, in which chemical structures are the basis of the development of new therapies. Cancer is considered one of the most important causes of mortality in both developing and developed countries [72]. Among plant hormones, 24-epiBL has a similar structure to animal steroidal hormones [73]. BRs not only exist in plants but rather in human cells, although scientific evidence is very scarce, and in both animals and plants, BRs regulate cell expansion and proliferation, but in animals, it can act on distant tissues after transport and circulation. During the last 10 years, BR derivatives have been checked for their antiproliferative and anticancer activities, although scientific evidence is very scarce. Various studies have shown the potential valuable activities of natural BRs for medical applications, including immunomodulatory, antiviral, antiproliferative, or neuroprotective activities [74,75,76].

BRs, at micromolar concentrations, have been reported to have effects on reducing cell proliferation and inducing apoptosis in different cancer cell types (breast, lung, prostate, and cervical cancer, among others) without affecting the cell growth and activity of normal cells, indicating tumour cell specificity [73,74]. Thus, small-cell lung cancer (SCLC) is a carcinoma accounting for 13% of all lung cancers, which is very aggressive, with rapid spread [73]. SCLC treated with 0.5 μM castasterone, which is converted to 24-epiBL, interacts with the cells and lowers the content of β-catenin, which is an indicator of the cancer, while in control samples, the content remained constant. In hepatocarcinoma, 24-epiBL has shown anticancer activity, although its mechanism remains unknown. In this report, 24-epiBL induced apoptosis and provoked energy restriction in the cancer [77].

Finally, it is worth mentioning that the metabolism and degradation of 24-epiBL and other plant BRs are sufficient in mammalian animals, including human beings, and the formation of compounds of potential concern for consumers or the environment has not been indicated. Thus, toxicological reference values for dietary risk assessment are considered to be not necessary [78].

## 7. Conclusions and Future Perspectives

This review presents for the first time the knowledge developed in the last decade on the role of BRs in fruit crop yield and quality properties, from preharvest to postharvest, which have been performed mainly in non-climacteric fruits. It has been proven that exogenous application of BRs in non-climacteric fruits has a high impact on development and compositional changes associated with ripening, apart from increasing tolerance to biotic stress, with the most important being CI. In this review, it was found that the most widely used BR analogue in research is 24-epiBL, with doses between 0.1 and 15 μM. Among the different functions of BRs in non-climacteric fruits, the following stand out: (a) signalling function in the regulation of fruit development through the induction of gene expression involved in cell division; (b) increase in crop yield and higher production in kg tree^−1^; (c) function of advancing or delaying ripening depending on the application dose, which is very important to stimulate the red colour by increasing anthocyanins; and (d) function of advancing or delaying senescence through modulating antioxidant enzymes. Finally, as future perspectives, it is proposed to extend to other non-climacteric fruits and to deepen our knowledge of how BRs regulate physiological aspects from preharvest to postharvest. It is also necessary to study the role of BRs against rot and biotic stress.

## Figures and Tables

**Figure 1 plants-13-03082-f001:**
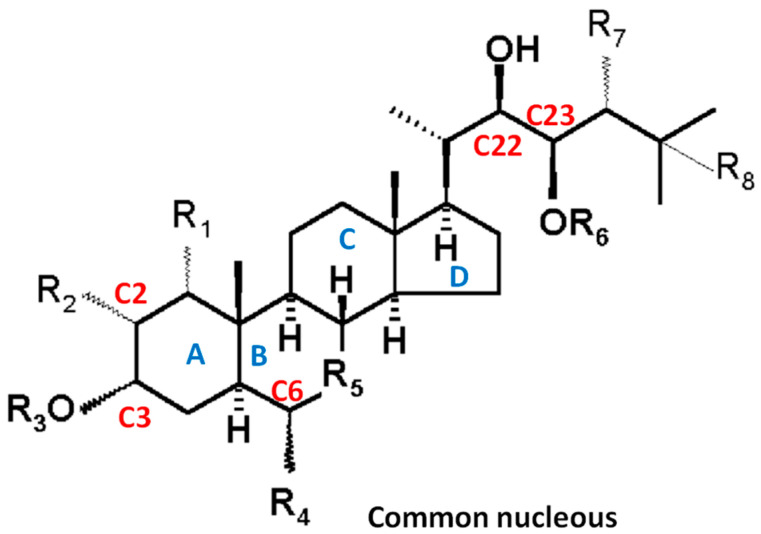
The common structure of the brassinosteroid (BR) family.

**Figure 2 plants-13-03082-f002:**
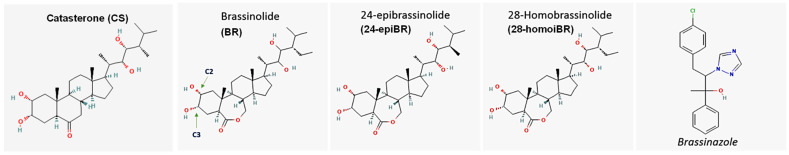
The chemical structure of the most important brassinosteroids (BRs) in plants and the inhibitor named ‘brassinazole’.

**Figure 3 plants-13-03082-f003:**
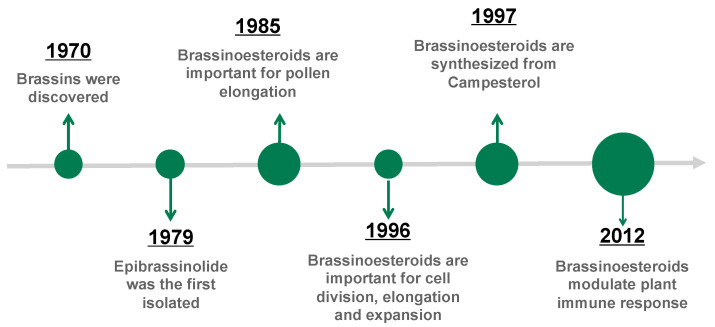
Brief story about important facts in brassinosteroid (BR) discovery. Size of green circles shows the importance of the discovering facts.

**Figure 4 plants-13-03082-f004:**
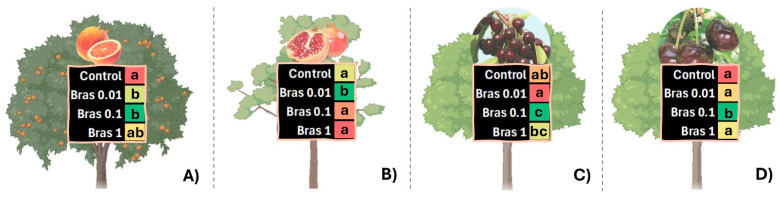
The effect of control and 24-epiBRL (Bras 0.01, 0.1, and 1 µM) foliar spray treatments on the crop yield of blood orange, cv. ‘Sanguinelli’ (**A**), pomegranate, cv. ‘Mollar de Elche’ (**B**), sweet cherries, cv. ‘Sunburst’ (**C**), and ‘Skeena’ (**D**). The colours red, orange, yellow, and green on the heatmap represent the range of crop yield values for each species and cultivar, from the lowest to the highest. These data are based on three replicates of three trees per treatment and cultivar. A completely randomised tree block design was used in the study. Different letters show significant differences (*p* < 0.05) among treatments according to Tukey’s test.

**Table 1 plants-13-03082-t001:** Preharvest application of brassinosteroids (BRs) in different fruit crops ^1^.

Common Name	Plant/Stage	BR Type	Doses	Reference
Sugar apple	Anthesis	24-epiBL	2.1 μmol L^−1^	[43]
Strawberry	Flowering	BR	0.1–0.3 mg L^−1^	[44]
Sweet cherry	Fruit growth	HBL	2 μmol L^−1^	[36]
Cantaloupe	Flowering	24-epiBL	0.2 μmol L^−1^	[45]
Table grape	Fruit growth	24-epiBL	4.2 μmol L^−1^	[46]
Strawberry	Flowering	BB16	0.1 mg L^−1^	[22]
Table grape	Veraison	24-epiBL	3–6 μmol L^−1^	[47]
Pepper	Flowering	24-epiBL	10^−6^ M	[48]

^1^ Abbreviations; BR: brassinolide; 24-epiBL: 24-epibrassinolide; HBL: 28-homobrassinolide.

**Table 2 plants-13-03082-t002:** Effects of brassinosteroid (BR) preharvest application on quality traits of different fruit species at harvest ^1^.

Fruit	Trait	Control	Treatment	Reference
Persimmon	Firmness (N)	40.0	50.0 *	[61]
	TSS (g 100 g^−1^)	17.0	16.4	
Table Grape cv. ‘Redglobe’	TSS (g 100 g^−1^)	16.7	16.5	[55]
	TA (g L^−1^)	7.8	7.4 *	
Table Grape cv. ‘Thompson Seedless’	TSS (g 100 g^−1^)	18.5	22.5 *	[47]
	TA (g L^−1^)	6.45	6.75 *	
Strawberry	TSS/TA Ratio	5.5	15.2 *	[44]
Apples cv. ‘Red Delicious’	Firmness (N)	4.5	5.9 *	[51]
	TSS (g 100 g^−1^)	14.5	15.2 *	
	TA (g 100 g^−1^)	0.3	0.2	

^1^ Abbreviations: TSS: total soluble solids; TA: titratable acidity. The symbol of * shows significant differences between the control and the treatment with BRs.

**Table 3 plants-13-03082-t003:** Effects of 24-epiBR preharvest treatments on bioactive fruit content at harvest. * shows significant differences between control and BR-treated fruits.

Fruit	BR	Trait	Control	Treated	Reference
Wine Grape	0.4 mg/L 24-epiBR	Anthocyanin	0.21 mg/g	0.86 mg/g *	[62]
Table Grape	3–6 μmol L^−1^ 24-epiBR	Ascorbic Acid	70 mg/100 g	100 mg/100 g	[47]
Strawberry	1 μM 24-epiBR	Vitamin C	45 mg 100 g^−1^	125 g 100 g^−1^ *	[56]
‘Red Delicious’ Apples	1 μM 24-epiBR	Anthocyanin Vitamin C Phenolics	5 g kg^−1^ 0.8 g kg^−1^ 23 g kg^−1^	6.5 g kg^−1^ * 1.3 g kg^−1^ * 32 g kg^−1^ *	[51]

**Table 4 plants-13-03082-t004:** Effects of postharvest BRs treatments in some fruit quality traits during postharvest storage. * shows significant differences between control and BR-treated fruits. All concentrations are expressed as fresh weight basis (FW).

Fruit	24-epiBL	Trait	Control	Treated	Reference
Zucchini	0.1 μM^−1^	CI Index Weight Loss Phenolics	2.9 5.9 520	2.2 * 3.2 * 640 *	[67]
Wine grape		Anthocyanin	0.3	0.5 *	[55]
Blood orange	8 μL L^−1^	Firmness TA Phenolics	1.1 20	3.5 * 32 *	[69]
Mandarin	5 mgL^−1^	Decay (%) Catalase Peroxidase H_2_O_2_	19 80 3485 0.31	5 * 145 * 4052 * 0.40 *	[68]
Pomegranate	15 μM	CI Index Decay (%) EL (%)	3.2 80 82	1 * 30 * 32 *	[70]
Strawberry	1 μM^−1^	Vitamin C Anthocyanin	45	125 *	[56]
Blood orange	10 mM	Anthocyanin EL (%) TA	100 82 13.4	240 * 65 * 21.3 *	[71]

CI: chilling injury; weight loss: %; phenolics: mg kg^−1^; anthocyanin: mg g^−1^; firmness: kg cm^−2^; H_2_O_2_: mmol g^−1^; catalase and peroxidase: U g^−1^ min^−1^; TA: titratable acidity g kg^−1^; vitamin C: g 100 g^−1^

## Data Availability

All data are provided in the manuscript.

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
