# Peer review of "Brassinosteroids: An Innovative Compound Family That Could Affect the Growth, Ripening, Quality, and Postharvest Storage of Fleshy Fruits"

_plants, 2024, doi:10.3390/plants13213082_

Round 1
Reviewer 1 Report
Comments and Suggestions for Authors
This review topic is interesting, the brassinosteroids is a relative new plant hormone that possess multiple effect, so this review can conclude its regulative effects. This paper have flaws that need improve, details are listed below.
1. Introduction of this review is too long-winded, because the BRs showed at the end of introduction.
2. line 92, do any knowledge about the metabolism of BRs, including biosynthesis and biodegradation.
3. the title of this paper mentioned focus on non-climacteric fruits, is there any differences between responses of non-climacteric and climacteric fruit to BRs? In table 1, the apple seem to be a kind of climacteric fruit, which should not involved in this paper.
4. line 392, as to its safety, is there any limit to fresh foods or threshold value of BRs to human?
Author Response
I added a file

Reviewer 2 Report
Comments and Suggestions for Authors
The review manuscript "Brassinosteroids: An innovative tool affecting fruit growth, ripening, quality and postharvest storage of non-climacteric fleshy fruits" is interesting readding and of social importance. I consider approved after minor revisions.
Considerations:
- In the title of review to: "Brassinosteroids: A compounds class that could innovate the affecting fruit growth, ripening, quality and postharvest storage of non-climacteric fleshy fruits", or then "compounds family". Anyway, I would change the word "tool".
- In the sentence "Bishop and Yokota [28] proposed defining BRs as steroids that have one oxygen at the C-3 carbon atom and additional oxygens at C-2, C-6, C-22 and C-23 (in accordance with the numerical sequence of the steroid carbons, as illustrated in Figure 1)" (lines 107-109) is stated that there is a numerical sequence of chemical structure of BRs; however, not all counting is done, missing to show C-6, C-22, and C-23. Please, it is possible add all the counting.
- In Figure 2 (or new figure) you could add chemical structures of compounds mentioned in the text: 5α-cholestane, 5α-ergostane, 5α-stigmastan, and catasterone and number them (in text and figure). Perhaps, to simplify could be used legend of radicals (R1, R2, R3,...) and/or of non-defined stereochemistry indicating in legend "hashed wedged" and "solid wedged" of each molecule.
Minor correction:
- Delete "such as" (line 69);
- Exclude the hyphen of "to-enhance" (line 80);
- Add a comma after C in "(A, B, C and D)" (line 93);
- In Figure 1 R3 is half erased. Please, change;
- Add a comma after C-22 in "C-2, C-6, C-22 and C-23" (line 108);
- In Figure 2, I suggest add abbreviations of Brassinolide, 24-epibrassinolide, and 28-homobrassinolide next to the names;
- Change "polle" to "pollen" (line 160);
- The word "on" is crossed out. Correct, please. (line 194);
- Add a comma after "cereals" (line 196);
- Please, specify "TSS/TA ratio", it is the first time that appears in the text;
- Change BRL to BRs or then specify BRL (line 236);
- Specify "UFS/SFA" and MDA (lines 380 and 384)
Author Response
I added a file
